# E2F4DN Transgenic Mice: A Tool for the Evaluation of E2F4 as a Therapeutic Target in Neuropathology and Brain Aging

**DOI:** 10.3390/ijms232012093

**Published:** 2022-10-11

**Authors:** Morgan Ramón-Landreau, Cristina Sánchez-Puelles, Noelia López-Sánchez, Anna Lozano-Ureña, Aina M. Llabrés-Mas, José M. Frade

**Affiliations:** 1Department of Molecular, Cellular and Developmental Neurobiology, Cajal Institute, Consejo Superior de Investigaciones Científicas, 28002 Madrid, Spain; 2Cajal International Neuroscience Center, Consejo Superior de Investigaciones Científicas, UAH Science and Technology Campus, Avenida León 1, 28805 Alcalá de Henares, Spain

**Keywords:** acetylated E2F4, synapsis, tissue homeostasis, Alzheimer’s disease, 5xFAD mice, neuroinflammation, microgliosis, reactive astrocytes

## Abstract

E2F4 was initially described as a transcription factor with a key function in the regulation of cell quiescence. Nevertheless, a number of recent studies have established that E2F4 can also play a relevant role in cell and tissue homeostasis, as well as tissue regeneration. For these non-canonical functions, E2F4 can also act in the cytoplasm, where it is able to interact with many homeostatic and synaptic regulators. Since E2F4 is expressed in the nervous system, it may fulfill a crucial role in brain function and homeostasis, being a promising multifactorial target for neurodegenerative diseases and brain aging. The regulation of E2F4 is complex, as it can be chemically modified through acetylation, from which we present evidence in the brain, as well as methylation, and phosphorylation. The phosphorylation of E2F4 within a conserved threonine motif induces cell cycle re-entry in neurons, while a dominant negative form of E2F4 (E2F4DN), in which the conserved threonines have been substituted by alanines, has been shown to act as a multifactorial therapeutic agent for Alzheimer’s disease (AD). We generated transgenic mice neuronally expressing E2F4DN. We have recently shown using this mouse strain that expression of E2F4DN in 5xFAD mice, a known murine model of AD, improved cognitive function, reduced neuronal tetraploidization, and induced a transcriptional program consistent with modulation of amyloid-β (Aβ) peptide proteostasis and brain homeostasis recovery. 5xFAD/E2F4DN mice also showed reduced microgliosis and astrogliosis in both the cerebral cortex and hippocampus at 3-6 months of age. Here, we analyzed the immune response in 1 year-old 5xFAD/E2F4DN mice, concluding that reduced microgliosis and astrogliosis is maintained at this late stage. In addition, the expression of E2F4DN also reduced age-associated microgliosis in wild-type mice, thus stressing its role as a brain homeostatic agent. We conclude that E2F4DN transgenic mice represent a promising tool for the evaluation of E2F4 as a therapeutic target in neuropathology and brain aging.

## 1. Introduction

E2 factor 4 (E2F4) is a member of the E2F family of transcription factors [1], which are primarily known to regulate the cell cycle. E2F4 was first described as a cell cycle repressor able to interact with p107 [2,3] and p130 [4], two members of the retinoblastoma (RB) family. Nevertheless, its capacity to repress cell cycle progression can be modulated, as it can also facilitate the cell cycle progression of cardiomyocytes, normal intestinal crypt cells, and colorectal cancer cells [5,6]. A number of reviews have been published describing the role of this transcription factor in quiescence and other cell cycle-related mechanisms [7,8,9], and we refer to the reader to these informative reviews for this aspect of E2F4 function.

Interestingly, E2F4 can also play other important roles in cellular physiology, including cell and tissue homeostasis and tissue regeneration [7,8,10,11,12]. Therefore, E2F4 can be considered a multifactorial factor with an important impact on neuronal welfare and brain homeostasis [11,12], suggesting that it may be a promising candidate target for neurodegenerative diseases and brain aging.

E2F4 is a phosphoprotein whose phosphorylation within an evolutionary-conserved threonine motif containing T248 (Figure 1) can modify its function [11,13]. This covalent modification has been targeted by substituting T248 and T250 with alanines, thus resulting in a dominant negative form of E2F4 (E2F4DN). This mutant form, or E2F4, prevents cell cycle re-entry in developing neurons [13] and is able to prevent Alzheimer’s disease (AD)-deleterious processes in 5xFAD mice [11], a murine model of this disease [14].

In this review, we will focus on the novel functions of E2F4 and their regulation as well as the covalent modifications of E2F4 that may modulate its function. We will also describe what has been published on E2F4DN transgenic mice, a mouse model generated in our laboratory that has been useful for the analysis of E2F4 as a multifactorial therapeutic factor for AD. Finally, we will describe how neuronal expression of E2F4DN reduces the neuroinflammatory response in both 5xFAD/E2F4DN double transgenic and wild-type (WT) mice at 1 year of age (i.e., middle-aged mice [15]).

## 2. Transcriptional and Non-Transcriptional Functions of E2F4

### 2.1. E2F4 as a Transcriptional Regulator

Human E2F4 contains 413 amino acids (410 in mouse) distributed throughout four domains (Figure 1). As with other E2F members, it forms a heterodimer with dimerization partner (DP) proteins through its dimerization domain (DD), located at the N-terminus of the molecule. The DD domain is required for its interaction with DNA through the DNA-binding domain (DBD). A third domain located at the C-terminus is required for the function of E2F4 as a transcription factor [16]; this transactivation domain (TD) is blocked when the retinoblastoma (RB) family proteins p107 or p130 interact with E2F4 through its protein-binding domain [10]. This interaction is crucial for the control of the E2F4 function as a transcription factor. Finally, E2F4 has a region that has been proposed as a regulatory domain (RD) [13] in which phosphorylatable residues, such as T248 (see below), are placed. In addition, two nuclear export signals (NES) are present in E2F4, one located within the DBD and the other in the DD [17,18]. These sequences maintain E2F4 within the cytoplasm unless it interacts with p107 or p130, which are required for the translocation of E2F4 to the nucleus. In addition, other factors can induce the translocation of E2F4 to the nucleus, as the latter can also regulate transcription through RB-independent mechanisms [19]. In this regard, E2F4 can interact with KPNB1, RanGAP1, and RanBP2 [19], three proteins that are involved in nuclear import [20,21], and may facilitate E2F4 nuclear translocation in the absence of RB family members. Moreover, E2F4 may be translocated to the nucleus with the help of DP family members DP-2 and DP-3 [22,23,24], likely due to the presence of a nuclear localization signal (NLS) in their sequence, as has been shown in DP-2 [25]. Finally, E2F4 harbors a weak putative NLS in amino acids 52-61 [25], suggesting that E2F4 can translocate into the nucleus in a cofactor-independent manner, similar to E2F5 during keratinocyte differentiation [26]. 

A ChIP-seq analysis performed in human lymphoblastoid cells identified around 16,000 E2F4 binding sites that potentially regulate 7346 target genes with wide-ranging functions, including cell cycle regulation, DNA repair, RNA processing, stress response, apoptosis, ubiquitination, protein transport and targeting, protein folding, and I-κB kinase/NF-κB cascade regulation [27]. In these cells, E2F4 can bind near transcription start sites (TSSs), a finding confirmed by others [28]. In addition, functional distal sites for E2F4 can be located more than 20 kb away from the annotated TSSs. In both cases, E2F4 can act as an activator as well as a repressor [27]. This analysis also indicated that E2F4 can bind to the promoters of 780 transcription factors, suggesting that E2F4 can indirectly regulate broad classes of genes [27]. Other authors have confirmed that E2F4 can bind to genes related to DNA repair, DNA damage, and G2/M checkpoints, as well as to other classical functions, such as cell cycle regulation, DNA replication, chromosome transactions, and mitotic regulation [29]. In most cases, E2F4 can bind to a specific promoter together with other members of the E2F family [28], indicating that the E2F4 function is subjected to complex cross-regulatory networks [30,31]. Many E2F4 binding sites have been analyzed in specific gene regulatory regions [32]. For instance, the release of a p130-E2F4 complex from sequences immediately upstream of the transcription initiation site of the human *CDC2* promoter has been shown to coincide with the induction of CDC2 expression [33]. 

Several lines of evidence indicate that E2F4 is able to control complex transcriptional regulatory networks in specific cells, thus supporting its multifactorial capacity as a transcription factor. For instance, a combined analysis using gene ontology and expression data has been used to define the network controlled by E2F4 in B cells [34]. In addition, loss-of-function studies on E2F4 silencing using a specific shRNA in acute myeloid leukemia cells have revealed that 276 genes show altered expression patterns in these cells [35]. These E2F4-regulated genes are mostly involved in the regulation of the mitogen-activated protein kinase (MAPK) signaling pathway. 

The regulation of gene transcription by E2F4 seems to be mediated through histone acetylation, as E2F4 may interact with CREB binding protein (histone acetyltransferase) [19], and sites where E2F4 binds are histone-modified [27].

### 2.2. E2F4 and Non-Transcriptional Interactors

E2F4 lacks a strong NLS, which suggests that this protein could play a significant role in the cytoplasm [36]. This is, for instance, the case of the regulation of centriole amplification during multiciliogenesis, which is mediated by the interaction of E2F4 with Deup1 and SAS6, two components of the centriole replication machinery [37]. Indeed, cytoplasmic E2F4 forms organizing centers in multiciliated cells [38]. While centrioles are known to undergo one round of duplication per cell cycle in normal proliferating cells, multiciliated cells show a massive assembly of these organelles during G0, a process initiated by Multicilin in combination with E2F4 (or E2F5) and Dp1 [39,40,41].

The capacity of E2F4 to function out of the nucleus is consistent with a study by Hsu and collaborators [19]. These authors identified a number of E2F4 interactors in mouse embryonic stem cells (mESCs) and a retinal pigment epithelium (RPE)-derived cell line of human origin [19]. Several of these interactors are located outside of the cell nucleus since a cellular component (CC) ontology analysis performed by us using the E2F4 interactors described by Hsu and collaborators [19] confirmed that E2F4 may be functional in the cytoplasm of mESCs (Appendix A) and both cytoplasm and extracellular vesicles from RPE-derived cells (Appendix A).

## 3. Regulation of E2F4 Function by Chemical Modifications

Proteins can be posttranslationally modified through covalent processing events that change their properties, either by proteolytic cleavage or by adding a modifying group, such as acetyl, phosphoryl, glycosyl, and methyl, to one or more amino acids [42]. More than 400 different types of posttranslational modifications [43] affect many aspects of protein function. Some of these chemical modifications have been described in E2F4.

As in the case of other regulators of the cell cycle, E2F4 can be ubiquitinated as a mechanism regulating its protein levels [44]. In addition, E2F4 activity could be modulated by protein acetylation, as observed with another member of the E2F family of transcription factors, E2F1 [45]. E2F1 can be acetylated in sites that lie adjacent to the DBD, thus increasing its DNA-binding ability and activation potential, as well as its protein half-life [45]. In the case of E2F4, Hsu and collaborators [19] demonstrated that both human and mouse E2F4 can be significantly acetylated in K37 and K96. These residues are located within the DBD and DD, respectively, thus suggesting that the capacity for DNA binding and DP heterodimerization of E2F4 can be compromised. This may facilitate the cytoplasmic function of E2F4 as a multifactorial protein. These authors also found small levels of acetylation in K20, K28, K44, K73, K82, K101, K177, K197, K230, and K347 from human E2F4 and in K28, K44, K101, K118, K177, K178, and K339 from mouse E2F4. Most of these residues are located within the DBD and DD of E2F4, suggesting that their acetylation can also participate in the regulation of DNA binding and the DP heterodimerization of E2F4.

Using an acetylated K96-specific antibody, we verified that K96 becomes acetylated in some structures of the adult mouse brain in vivo (Figure 2). This form of acetylated E2F4 can be detected in NeuN-positive cells (i.e., neurons) within the hippocampus (dentate gyrus) (Figure 2a), cerebellum (Figure 2b), and NeuN-negative cells located in the rostral migratory stream (RMS) (Figure 2c), likely neural progenitors. Some NeuN-negative cells in the cerebellum also showed acetylated E2F4-specific immunoreactivity (Figure 2b).

In non-histone proteins, methylation represents a chemical modification participating in diverse processes, such as cell cycle control, DNA repair, senescence, differentiation, apoptosis, and tumorigenesis [46]. As a multifactorial factor, E2F4 can also become methylated. In this regard, Hsu and collaborators [19] have shown that a significant proportion of K73, K197, and R357 (R360 in mice) residues from E2F4 can be methylated. Interestingly, the methylation of K197 in E2F4 is reminiscent of a similar process in E2F1, affecting K185, which is involved in the regulation of E2F1-induced cell death [46,47,48]. Other residues of human (K20, K37, K53, K57, K74, K96, K101, R147, K177, K230, and K347) and mouse (R297 and K339) E2F4 can also be methylated, as reported by Hsu and collaborators [19].

Finally, the most prominent mechanism regulating E2F4 activity is protein phosphorylation. E2F4 has several residues susceptible to phosphorylation (Figure 1), and several lines of evidence indicate that E2F4 can undergo phosphorylation [49] to modulate its function. In this regard, this chemical modification may regulate E2F4-mediated transcription, either by disrupting its DNA-binding ability, as observed in 3T3 cells [50], or by enhancing the DNA binding of the E2F4/p130 repressor complex, as demonstrated in human fibroblasts [51]. Seven of the theoretical phosphorylation sites of E2F4, including T14, S202, S218, T224, S244, T248, and S384, have been demonstrated to become phosphorylated [52]. Other authors have confirmed the phosphorylation of T14, S218, S244, T248, and S381 in human E2F4 [19], of S218, T224, T249, and S384 in mouse E2F4 [19], and the ortologue of T248/T250 (T261/T263) in chicken E2F4 [13]. In addition, phosphorylation of E2F4 in T249 has been observed in mouse brain extracts using a phosphosite-specific antibody [11], and indirect evidence for the phosphorylation of T248 in the human brain was obtained using a proximity ligation assay with anti-E2F4 and anti-phosphothreonine antibodies [12]. Hsu and collaborators [19] also found evidence of phosphorylation in S16, Y139, S185, S187, S220, S223, and Y389 from human E2F4 and in S75, Y139, T153, S223, S240, T266, Y392, and Y394 from mouse E2F4.

We will further discuss the effects of E2F4 phosphorylation in Section 5.2.

## 4. E2F4 as a Multifactorial Regulator

### 4.1. E2F4 as a Regulator of Tissue Homeostasis

In addition to its classical function in regulating quiescence in proliferating cells, E2F4 can also participate in several homeostatic processes. For instance, E2F4 has been associated with the DNA damage checkpoint and repair pathways [29,53,54] (see below), prevention of DNA damage-associated cell death [31], repression of apoptotic genes [55], epigenetics [56], metabolism regulation [57,58], autophagy [59], inflammation [60], and cell repair [61]. In addition, E2F4 function has been associated with oxidative stress [62]. In this regard, the p107-E2F4 complex downregulates PGC-1alpha expression [63], an enzyme that protects cells against oxidative stress and reduces mitochondrial dysfunction in AD [64,65].

The ability of E2F4 to regulate several homeostatic functions may have evolved from its capacity to regulate processes primarily associated with cell cycle arrest and cell differentiation. Indeed, under growth arrest conditions, E2F4 can repress a common set of genes involved in mitochondrial biogenesis and metabolism [66]. Moreover, E2F4 participates in the differentiation of multiple cell types, including the differentiation of myocytes [22,36,67,68,69], neural cells [30,70], adipocytes [71,72,73,74], hematopoietic cells [75], chondrocytes [76], extra-embryonic tissues [77], endothelial cells [78], epithelial cells [79], and multiciliated cells [80,81]. E2F4 can also regulate eye and brain patterning [82,83,84,85], as well as endocytosis and water channel transport in the testes [81].

The capacity of E2F4 to act as a multifactorial factor is likely mediated by the different interactors to which this molecule can bind. In this regard, E2F4 can perform non-canonical actions in cells in the absence of RB family proteins, allowing the transactivation domain to interact with other proteins [19]. After performing biological process (BP) ontology analysis, we found that many E2F4 interactors identified by these authors are related to non-cell cycle processes, including DNA repair, stem cell population maintenance, protein sumoylation in mESCs (Appendix A), as well as retina homeostasis, RNA splicing, organ regeneration, and regulation of lipid kinase activity in RPE-derived cells (Appendix A).

### 4.2. E2F4 as a Regulator of DNA Repair

Cells have to constantly respond to genotoxic insults that may induce DNA modifications, which usually lead to genome instability. Accumulation of damaged DNA is deleterious for cells since it often results in abnormal proliferation, cell aging, or cell death. Eukaryotic cells have acquired mechanisms of defense against this damage; globally, they are referred to as DNA damage response (DDR), which are in charge of monitoring and removing lesions in their DNA [86]. In this regard, mammalian cells are equipped with several DNA repair pathways, which can be classified into two main groups [87]. On the one hand, the machinery involved in base excision repair, nucleotide excision repair (NER), and mismatch repair can fix single-strand mutations. On the other hand, double strand breaks (DSBs) can be repared through two main mechanisms: homologous recombination (HR), which repairs DSBs during the S-phase or G2 since the sister chromatic is used as a template, and non-homologous end-joining (NHEJ), which is able to repair DSBs at any stage of the cell cycle and in quiescent and postmitotic cells.

DDR can be transcriptionally regulated by E2F factors. These transcription factors usually bind to two adjacent E2F sites within the regulatory regions of genes involved in DNA damage checkpoint and repair [88], thus allowing for functional interactions. Two known E2F factors regulating DDR are E2F4 and E2F1 [27,29], which functionally counteract each other. For instance, E2F4 silencing in MCF7 epithelial breast cells treated with benzoapyrene, an environmental pollutant that triggers DNA damage [89], results in E2F1 derepression and the subsequent induction of DNA repair factors [90]. In primary neurons, the repair response to DSBs is also regulated by E2F1 and E2F4. In this cellular system, E2F1 enhances *Cited2* expression, a pro-apoptotic gene required for delayed neuronal cell death, while E2F4 strongly inhibits *Cited2* transcription, helping to cell survival [31]. Finally, E2F4 has also been implicated in NER since the p130/E2F4 complex controls the expression of xeroderma pigmentosum complementation group C [53], a protein that serves as the primary initiating factor in the global genome NER pathway [91]. There is also evidence that hypoxia and the anti-angiogenic agent cediranib are both able to induce the binding of p130/E2F4 complexes to E2F consensus sequences in the promoters of homology-directed DNA repair genes, thus reducing gene expression [54,88,92].

In most paradigms, E2F4 seems to act as a repressor of genes involved in DNA damage checkpoint and repair. This function may be favored by the stress kinase p38^MAPK^, which phosphorylates E2F4 [13] and becomes activated by the DDR [93]. Therefore, the expression of a non-phosphorylatable form of E2F4 (E2F4DN) might modulate the maintenance of the expression of genes involved in DDR.

### 4.3. E2F4 as a Putative Regulator of Synaptic Function

E2F4 has been related to cognitive impairment [94] and the pathogenesis of AD [95], as well as to other neurological diseases [96], including Parkinson´s disease/mild cognitive impairment [97]. Since AD is largely a synaptic failure [98] occurring prior to cognitive decline or cell death [99], it can be speculated that E2F4 is important for synaptic function.

#### 4.3.1. Transcriptional Regulation of Synaptic Function by E2F4

E2F4 has the potential to regulate the expression of an ample number of synaptic proteins. As evidenced by ChIP-seq datasets from the ENCODE transcription factor targets dataset interrogated with the Harmonizome tool [100], E2F4 can bind to 46 synaptic protein-encoding genes (Appendix A), as well as 127 genes encoding for ion channel subunits (Appendix A). In this regard, there is direct evidence that E2F4 can regulate synaptic function, coming from the transcriptomic analysis performed in mESCs subjected to E2f4 gene knock-out (KO) (see genes included in Appendix A from the study by Hsu and collaborators [19]). The transcriptional alterations in synaptic plasticity-related genes upon E2F4 modulation reveal the potential role of this protein in synaptic function. This suggests that E2F4 could be a promising target for several neurological diseases that course with synaptic plasticity impairment, such as AD.

#### 4.3.2. Interaction of E2F4 with Synaptic Regulators

E2F4 can interact with synaptic regulators. We verified using BP ontology that almost half of the E2F4 interactors found in the study by Hsu and collaborators [19], which are common in both mESCs and RPE-derived cells, have a function in either axonal transport or synapse physiology (Appendix A).

The E2F4 interactors involved in synaptic function that were identified in RPE-derived cells include Rac Family Small GTPase 1 (Rac1), cell division cycle 42 (Cdc42), and protein phosphatase 1 catalytic subunit β (PPP1CB) [19]. The actin regulators Rac1 and Cdc42 are important for the structural and functional plasticity of dendritic spines, which are the basis of learning mechanisms [101]. The actin cytoskeleton regulator Rac1 controls synaptic actin dynamics [102] and is involved in actin-regulated short-term presynaptic plasticity through the modulation of synaptic vesicle replenishment [103]. Cdc42 is known to have an important role in dendritic branching [104], and it is part of the mechanism involved in CaMKII activation, which modulates dendritic spine structural plasticity and induces LTP [105]. PPP1CB is one of the three catalytic subunits of protein phosphatase 1 (PP1), a serine/threonine protein phosphatase that regulates synaptic transmission and plasticity [106]. PP1 mediates NMDAR dephosphorylation, modulating the synaptic expression of this receptor [107].

Hsu and collaborators [19] also found Fragile X Mental Retardation Protein (FMRP) to be a candidate cofactor for E2F4 in mESCs. FMRP is an important regulator of activity-dependent plasticity in the brain, and the mutation in the FMR1 gene and subsequent loss of its protein product lead to Fragile X Syndrome (FXS), an inherited cause of autism and intellectual disability [108]. Mechanistically, FMRP is an RNA-binding protein that regulates the synthesis of synaptic and nuclear proteins within various compartments of the neuron [109]. FMRP binds to dendritic mRNA [110], and this may be important in mRNA localization to dendrites [111]. Thus, the hypothetical interaction of E2F4 with FMRP could be responsible for the modulation of synaptic protein transduction.

Hsu and collaborators [19] also found that Snapin, a protein related to synaptic function [112,113], can interact with E2F4 in both mESC and RPE cells.

In addition, the indirect effects of E2F4 on synaptic plasticity have also been described. In this regard, E2F4 can interact with Suv39H1 [114], a histone methyl transferase with an essential role in H3K9me3 methylation that mediates hippocampal memory functions [115].

The interaction of E2F4 with known synaptic regulators suggests that it may modulate synaptic function. This hypothesis is consistent with the observed enrichment of E2F1 in synaptic fractions, which is related to PSD95 expression and becomes upregulated with aging [116]. Furthermore, E2F1 is necessary for de novo neuronal tetraploidization occurring in mice, and this is associated with the alteration of cognition, as mice lacking this transcription factor show enhanced memory acquisition and consolidation [117]. Since E2F1 and E2F4 have antagonistic roles in neuronal function [96], we speculate that E2F4 could facilitate synaptic function and cognition, as opposed to E2F1.

#### 4.3.3. E2F4 and MAPK Proteins in Synaptic Function

Another piece of evidence for the putative capacity of E2F4 to regulate synaptic function comes from the study by [35], which showed that E2F4 can regulate genes involved in the MAPK signaling pathway. Although this pathway has been associated with cancer [35], it is also relevant for synaptic plasticity and AD [118,119,120]. A relevant member of the MAPK family of protein kinases is p38^MAPK^, the kinase that phosphorylates E2F4 in the Thr conserved motif controlling neuronal tetraploidization [13]. p38^MAPK^ is a protein involved in synaptic plasticity and memory impairment that has been widely related to AD [120,121]. Accordingly, p38^MAPK^ is progressively activated in neurons affected by AD [122] as well as in APP transgenic mice brains [121], and neuronal p38α^MAPK^ mediates synaptic and cognitive dysfunction in a murine model of AD by controlling amyloid-β (Aβ) production [120]. Moreover, downregulation in APP/Tau-transgenic mice of p38^MAPK^ results in the upregulation of genes involved in the MAPK pathway and calcium signaling [121]. Although the implication of E2F4 in this paradigm remains unclear, the expression of some calcium signaling and/or synaptic plasticity-related genes is altered upon p38α-MAPK deficiency in neuronal populations. In particular, the expression of both Grin2a and its encoded protein glutamate ionotropic receptor NMDA type subunit 2A (Grin2a) is decreased, resulting in a reduction of calcium influx in p38α-MAPK-deficient neurons [121]. Finally, knocking down *E2f4* using an *E2f4*-specific shRNA significantly decreased the protein levels of p-ERK [35], a key MAPK that has been involved in both neurodegenerative diseases, as well as in endocannabinoid [123,124,125,126,127,128] and calcium signaling [101,105,129,130,131,132,133], which are critical pathways in synaptic function and modulation.

## 5. E2F4 and Alzheimer’s Disease (AD)

AD is a neurodegenerative condition that represents the most common form of dementia. It is characterized by memory and cognitive impairment, which are typically present in the early stages of the disease. Further clinical outcomes include a decline in visuo-spatial skills and neuropsychiatric disorders (apathy, irritability, aggressivity, wandering, and hallucinations). In a lower percentage, other AD symptoms include difficulty or inability to perform activities, olfactory disorders, pyramidal and extrapyramidal motor signs, myoclonus, seizures, and sleep complications [134,135]. AD is classified into two types, early onset AD (EOAD) or familial AD [136] and late-onset AD (LOAD) or sporadic AD [136]. From a neuropathological point of view, this disease is characterized by the presence of amyloid plaques, neurofibrillary tangles, neuroinflammation, and neurodegeneration in the brain [137].

AD is an unmet need, without any approved cure or disease-modifying therapy. Current treatments are addressed to ameliorate symptoms. Pharmacological treatments have evolved in recent years and have been based on drugs for neuropsychiatric symptoms, including antipsychotic, anxiolytic, anti-depressant and anti-convulsant drugs [138].

### 5.1. AD Etiology

The etiology of AD is complex, and several hypotheses co-exist. The first descriptions of AD were based on the neuropathological phenotype of extracellular Aβ accumulation and neurofibrillary tangles, suggesting that Aβ processing was the upstream cause of AD [137,138].

Nevertheless, several studies support that Aβ processing abnormalities are necessary but not sufficient to lead to marked synaptic and neuronal loss [139]. Possible Aβ-independent mechanisms for AD etiology include synapse loss [140], altered glucose metabolism [141], cholesterol and lipid metabolism [138], oxidative stress [142], chronic hypoperfusion [143], cell adhesion pathways [138], immune system [138], and neuronal cell cycle re-entry [144] leading to tetraploidization [145].

The mutual interaction of all of these mechanisms makes it difficult to appropriately target the disease, and no effective therapies against AD are available until now. This is likely because the experimental therapies developed so far have mainly focused on single targets. Therefore, a paradigm shift is necessary, making it essential to design a multifactorial approach against this complex disease [146].

### 5.2. Connection of E2F4 with AD

As indicated above, E2F4 can regulate more than 7000 genes involved in several activities key to AD progression, such as DNA repair, RNA processing, stress response, apoptosis, ubiquitination, protein transport and targeting, protein folding, and I-κB kinase/NF- κB cascade, according to studies performed in a lymphoblastoid cell line [27]. As observed in this cell line, as well as in mouse embryonic stem cells, E2F4 may activate or repress gene expression according to its interaction partner [19,27]. Interestingly, E2F4 malfunction has been linked to cognitive impairment [94], as well as to the etiopathology of neurodegenerative diseases, such as Alzheimer’s disease (AD) [11,12]. This is consistent with a recent study that proposes E2F4 as a major regulator of most AD-specific gene networks [95], and with other bioinformatics-based studies suggesting that E2F4 participates in this disease [147,148,149]. Moreover, a genome-wide association study for late-onset AD has identified a single nucleotide polymorphism that modifies a DNA-binding motif of E2F4 as relevant for the disease [150].

As mentioned above, E2F4 can be phosphorylated at multiple residues, including T248 (T249 in the mouse sequence) [19]. In vitro studies in differentiating chicken retinal neurons have provided insight that phosphorylation of E2F4 is key for the expression of cell cycle progression genes. In this model, nerve growth factor (NGF) can activate neurotrophin receptor p75, which in turn induces nuclear p38^MAPK^ activity. As a result, E2F4 is phosphorylated in the T261/T263 motif, a change that allows cell cycle re-entry in these neurons [13], a mechanism generating neuronal tetraploidy [151].

We demonstrated in developing chick neurons that the expression of a dominant negative variant of chick E2F4 (E2F4DN) containing Ala substitutions in the Thr residues orthologous to T248 and T250 can prevent cell cycle re-entry in these cells and the subsequent DNA duplication that results in somatic neuronal tetraploidy [13]. Recent studies in our laboratory have confirmed that expression in the neurons of both mouse and human E2F4DN prevents neuronal tetraploidy in 5xFAD mice [12]. Moreover, as expected from a multifactorial factor, neuronal expression of E2F4DN was able to mitigate other processes that become affected in AD, such as neuroinflammation, Aβ peptide proteostasis, and body weight loss [12], a known somatic alteration associated with AD [152]. This results in the prevention of cognitive impairment [12]. Moreover, indirect evidence suggests that E2F4DN could also regulate synaptic function, as E2F4 has been shown to interact with a number of synaptic regulators in stem cells and in a photoreceptor-derived cell line (see above). All of these findings have allowed the development of an innovative gene therapeutic approach using human-derived E2F4DN [11] (see below).

Based on the above evidence, we postulate that E2F4 represents a potential multifactorial target for AD, as this transcription factor possesses an intrinsic capacity to modulate several processes that are affected by this disease, thus reestablishing brain homeostasis and favoring brain tissue regeneration [19]. The homeostatic capacity of E2F4 could be crucial in counteracting any physiological stress [153] associated with the etiology of AD. In this context, the phosphorylation of E2F4 in the conserved T248/T250 motif could alter its homeostatic function, which would be restored by E2F4DN expression [11,12].

#### 5.2.1. E2F4, Cell Cycle Re-Entry, and Neuronal Tetraploidy

Human studies performed with AD brain samples have demonstrated that neurons overexpress cell cycle markers, including S-phase regulators [154,155,156,157,158,159,160]. This suggests that cell cycle re-entry participates in the etiology of AD. According to this hypothesis, a number of studies in mice have revealed that forced cell cycle re-entry in response to oncogene expression leads to the neuropathological hallmarks of AD, including Tau phosphorylation and neurofibrillary tangles [161,162], extracellular Aβ deposits [161], gliosis [163,164], synaptic dysfunction [165], neuronal death [163,165], and cognitive deficits [163], reinforcing that this process participates in the etiology of AD. Furthermore, neuronal cell cycle re-entry in humanized Aβ plaque producing mice results in the development of additional AD-related pathologies, namely, pathological tau, neuroinflammation, brain leukocyte infiltration, DNA damage response, and neurodegeneration [166].

Once neurons re-enter the cell cycle, DNA is duplicated, but neurons are rarely observed to undergo mitosis [145]. As a consequence, tetraploid neurons are generated [167,168,169], and this process represents an early hallmark of AD [170,171] that precedes [169,170] and recapitulates [169] the neuropathology associated with this disease. The increase in tetraploid neurons in the AD preclinical stage might contribute to cognitive impairment and neuronal death susceptibility at late stages [170].

Recent in vitro studies have shown that hyperploidy impacts neuronal morphology [172] and causes both synaptic dysfunction and delayed neuronal death [165], as previously observed in AD-affected hyperploid neurons [170]. A decreased density of PSD95 puncta and reduced AIS length correlated with an alteration in synaptic function and excitability in these neurons. Furthermore, neuron hyperploidization leads to diminished action potential generation and reduced spontaneous synaptic activity, with lower amplitudes of synaptic events when compared to control cells [165]. Interestingly, membrane depolarization with high K^+^, which mimics electrical input, increases the survival of hyperploid neurons without reversing synaptic dysfunction. Therefore, it has been hypothesized that AD-associated tetraploid neurons could be sustained in vivo if integrated into active neuronal circuits while promoting synaptic dysfunction. As a result of this synaptic dysfunction and enhanced survival, silent tetraploid neurons disturb the network of neural circuits, leading to the neurological abnormalities observed in AD. In fact, in silico studies have concluded that neuronal tetraploidy could lead to major effects in AD through alterations in the firing frequency caused by neuronal network disruption [172]. Therefore, the relationship between cell cycle reactivation and AD neuropathogenesis may rely, at least partially, on the generation of tetraploid neurons.

Neuronal tetraploidy could also participate in the etiology of aging-dependent cognitive impairment, a process that takes place in individuals older than 40 years [173]. Indeed, a significant correlation between age and the proportion of tetraploid neurons was specifically observed in the entorhinal cortex of non-demented individuals [169], a known structure involved in memory formation [174]. In this context, age-associated neuronal tetraploidization can also be observed in the cerebral cortex of WT mice, while the blockade of neuronal tetraploidy in *E2f1*-deficient mice results in cognitive potentiation [169].

As indicated above, E2F4 controls cell cycle re-entry in neurons [13], and its expression becomes upregulated in cortical neurons from APP/PS1 mice [117]. A similar E2F4 upregulation is also observed in the prefrontal cortex of AD patients [144], as well as in neurons derived from human-induced pluripotent stem cells obtained from familial AD patients [148]. Furthermore, E2F4 becomes Thr phosphorylated in the cerebral cortex of APP/PS1 mice and Alzheimer’s patients [11,12,117]. As indicated above, phosphorylation of these two conserved Thr residues of E2F4 is necessary to induce neuronal tetraploidization and cognitive loss in AD, while expression of E2F4DN prevents these latter effects [11,12,117]. Therefore, E2F4 is a crucial agent regulating neuronal tetraploidization and its concomitant effects in the etiology of AD.

#### 5.2.2. E2F4 and Neuroinflammation, Aβ Peptide Proteostasis, and Body Weight Loss

Studies performed in our laboratory have demonstrated that E2F4 fulfils a multifactorial effect in AD, as the expression of E2F4DN in neurons attenuates microgliosis and astrogliosis, two hallmarks of neuroinflammation, modulates Aβ peptide proteostasis and prevents body weight loss in 5xFAD mice.

The paracrine effect of E2F4DN on neuroinflammation is likely mediated by either the cell membrane or extracellular factors released by E2F4DN-expressing neurons. Indeed, several mechanisms of bidirectional neuron–glia communication [175] have been described. In addition, neuron–glia communication can also take place through neuron-released exosomes [176].

The reduction of Aβ peptide levels in the hippocampus of 5xFAD mice in response to E2F4DN-based gene therapy [11] suggests the existence of a neuron-intrinsic capacity of the unphosphorylated form of E2F4 to prevent the production of this neurotoxic molecule. Nevertheless, the existence of a transcriptional program favoring Aβ peptide proteostasis in the double transgenic 5xFAD/E2F4DN mice suggests that E2F4DN may also induce cell-extrinsic effects on Aβ peptide proteostasis by acting on gene networks involved in processing, accumulation, and toxicity of Aβ [12].

E2F4DN expression in neurons can also reverse the loss of body weight observed in 5xFAD mice [11,12]. Since weight loss is likely associated with AD-associated metabolic alterations [177,178], the effect of E2F4DN on this trait may be due to a hypothetical capacity to affect neurons involved in sensing leptin [179], an adipocytokine that regulates energy metabolism and appetite [180]. E2F4 also has a connection with metabolic pathways since it can regulate insulin signaling in preadipocytes [74]. Therefore, E2F4 seems to participate in multiple pathways involved in energy metabolism and obesity, and this property may underline the capacity of E2F4DN to reverse weight loss in 5xFAD mice.

#### 5.2.3. E2F4 and Cognitive Impairment

Neuronal E2F4DN expression prevents the cognitive deficits observed in 5xFAD mice [11,12], suggesting that E2F4 phosphorylation in the conserved Thr motif prevents its effects on multiple pathways involved in cognition, thus resulting in cognitive loss. Many regulatory pathways may favor cognitive rescue by E2F4DN. First, evidence has accumulated during the last decades connecting the cell cycle with synaptic plasticity, as common molecules are involved in both processes [181]. Therefore, the capacity of E2F4DN to prevent cell cycle re-entry in neurons and the concomitant tetraploidization process could prevent synaptic dysfunction in affected neurons [165]. Furthermore, the hypothetical capacity of E2F4 to regulate synaptic plasticity (see above) could also participate in the recovery of cognition observed in 5xFAD mice expressing neuronal E2F4DN [11,12].

Second, neuroinflammation has an important impact on synaptic plasticity and memory. On the one hand, activated microglia secrete cytokines, chemokines, and reactive oxygen species, which can lead to synaptic plasticity and memory deficits [182]. On the other hand, synapses can be functionally altered when astrocytes become reactive, thus causing hippocampal circuit dysfunction and memory alterations [183]. Therefore, the capacity of E2F4DN to attenuate neuroinflammation in 5xFAD mice may also account for its beneficial effects on cognition.

Finally, the effects of neuronal E2F4DN expression on Aβ peptide proteostasis may favor cognitive recovery in 5xFAD mice, as this peptide is neurotoxic and known to trigger synaptic dysfunction and network disorganization [184].

## 6. E2F4DN Transgenic Mice and Neuroinflammation

To explore the therapeutic capacity of E2F4DN, we generated a knock-in (KI) mouse strain expressing mouse E2F4 with the T249A/T251A mutations (E2F4DN mice), Myc tagged at the C-terminus, and expressed under the control of the neuron-specific microtubule-associated protein tau (*Mapt*) promoter [12]. This transgenic mouse strain represents an optimal tool for the evaluation of E2F4 as a therapeutic target in neuropathology and brain aging. As a control, we used KI mice expressing EGFP under the *Mapt* promoter (EGFP mice) [185].

As mentioned above, hybrid mice resulting from the breeding of E2F4DN with 5xFAD mice (i.e., 5xFAD/E2F4DN mice) show a transcriptional program consistent with the attenuation of the immune response and brain homeostasis [12]. This correlates with the blocking of neuronal tetraploidization, the prevention of cognitive impairment, and the absence of body weight loss, a known somatic alteration associated with AD [152]. Consistently, 5xFAD/E2F4DN mice showed reduced microgliosis and astrogliosis at 3-6 months of age [12]. We further studied whether this effect is maintained at 1 year of age.

### 6.1. E2F4DN and Microgliosis

To verify whether microgliosis is reduced at 1 year of age in 5xFAD/E2F4DN mice compared with 5xFAD/EGFP mice, we crossed 5xFAD mice with either E2F4DN or control EGFP mice. Then, cortical sections of 1 year-old WT/EGFP, WT/E2F4DN, 5xFAD/EGFP and 5xFAD/E2F4DN mice were immunolabeled with Iba1, a specific microglia marker [186]. This analysis indicated that, as expected, the area occupied by microglia in the cerebral cortex of 5xFAD/EGFP mice was significantly greater than that of WT/EGFP mice (Figure 3a,b). This increase was associated with cortical layers 5–6 (Figure 3c). Therefore, as occurs at earlier time points [12], microglial cells are also activated in the cerebral cortex of 5xFAD/EGFP mice of 1 year of age.

As observed at earlier time points [12], the presence of E2F4DN significantly diminished the area occupied by microglial cells in 1-year-old 5xFAD mice (Figure 3a,b), further supporting the hypothesis that neuronal E2F4DN attenuates the microgliosis observed in 5xFAD mice. Interestingly, E2F4DN was also able to prevent an increase in the area occupied by microglial cells in the cerebral cortex of WT/E2F4DN when compared with WT/EGFP mice (Figure 3a,b), confirming what was observed at 6 months of age [12]. These effects were observed in all cortical layers, except in layer 6, where WT/E2F4DN mice showed a non-significant tendency to decrease the Iba1-occupied area when compared with WT/EGFP mice (Figure 3c). Therefore, the previously described age-dependent increase in microgliosis in the cerebral cortex [187,188] is prevented by our therapeutic molecule.

A significant reduction of the area occupied by microglia was also evident in the hippocampus of 1 year-old 5xFAD/E2F4DN mice when compared with 5xFAD/EGFP mice littermates of the same age (Figure 4). In addition, this same effect was observed in WT mice expressing E2F4DN (Figure 4), further supporting the hypothesis that the neuronal expression of our molecule can reverse the increase in microgliosis associated with brain aging in the hippocampus [187,188,189].

### 6.2. E2F4DN and Reactive Astrogliosis

As mentioned above, reactive astrogliosis is known to increase with age in the cerebral cortex of 5xFAD mice compared to WT mice [12]. To study the effect of E2F4DN on the reactive astrogliosis observed in mature 5xFAD mice, we crossed 5xFAD mice with either E2F4DN or control EGFP mice. Then, cortical sections of 1 year-old WT/EGFP, WT/E2F4DN, 5xFAD/EGFP and 5xFAD/E2F4DN mice were immunolabeled with the specific actrocytic marker GFAP [190]. This analysis demonstrated that, as occurs with microglial cells, the area occupied by GFAP immunoreactivity in the cerebral cortex of 5xFAD/EGFP mice is significantly greater than that of WT/EGFP mice (Figure 5a,b). This increase was associated with all cortical layers except layer 1 (Figure 5c). Therefore, as occurs at earlier time points [12], microglial cells are also activated in the cerebral cortex of 5xFAD/EGFP mice of 1 year of age.

As observed at 3 months [12], the presence of E2F4DN significantly diminished the area occupied by GFAP immunoreactivity in 1-year-old 5xFAD mice (Figure 5a,b), further supporting the hypothesis that neuronal E2F4DN expression attenuates the reactive astrogliosis observed in 5xFAD mice. This effect was observed in cortical layers 4 and 5 when 5xFAD/EGFP mice were compared with 5xFAD/E2F4DN (Figure 5c). This observation supports the hypothesis that the neuronal expression of E2F4DN can attenuate the increase in reactive astrocytes associated with AD. In contrast, E2F4DN was not able to reduce the area occupied by GFAP immunoreactivity in the cerebral cortex of WT/EGFP mice (Figure 5a,b), confirming what was observed at 3 months of age [12].

In the hippocampus, where GFAP was expressed by astrocytes at high basal levels in both WT and 5xFAD mice (Figure 6a), no difference was observed when E2F4DN was expressed in both WT/E2F4DN and 5xFAD/E2F4DN mice (Figure 6a,b).

## 7. Discussion

In this review, we have included experimental results on two novel aspects of E2F4 function. On the one hand, by using an acetylated K96-specific antibody, we provide immunohistochemical evidence that E2F4 can be acetylated in K96 in neurons and cells located within the RMS, thus confirming the finding by Hsu and collaborators [19] demonstrating the presence of K96-acetylated E2F4 in mESCs and an RPE-derived cell line. On the other hand, we analyzed at 1 year of age the neuroinflammatory state of double transgenic 5xFAD mice expressing E2F4DN in neurons. This analysis constitutes a follow-up of a previously published study performed in transgenic mice of 3 and 6 months of age.

Our results confirm the capacity of E2F4DN to attenuate microgliosis in the cerebral cortex and hippocampus of 5xFAD mice, even after one year, thus indicating that it has long-lasting therapeutic effects. In addition, E2F4DN was able to decrease the area occupied by GFAP cells (i.e., reactive astrocytes) in the cerebral cortex of the 5xFAD mice, while no changes in the area occupied by GFAP were observed in the hippocampus. This latter result contrasts with the observation that the area occupied by GFAP in the hippocampus of 3-month-old 5xFAD mice is decreased in the presence of neuronal expression of E2F4DN [12]. This discrepancy may be explained by the attenuation of astrocytosis in the hippocampus of 5xFAD mice at 1 year of age, a tissue where, in contrast to the cerebral cortex, GFAP is already expressed by non-reactive astrocytes. Therefore, E2F4DN-based gene therapy is likely to be effective in the long range. In this regard, we proved that our gene therapeutic approach is able to maintain the expression of E2F4DN for at least 1 year without major reduction in the transgene expression levels [11]. Since the durability of gene therapy has been reported to be good for years in humans [191], we expect that our E2F4DN-based gene therapy will require only one application for its effectiveness.

Our results are also in favor of the hypothesis that E2F4DN plays a role in preventing brain aging. This is evidenced by the capacity of our therapeutic protein to reduce the levels of microgliosis in the cerebral cortex and hippocampus of 1-year-old WT mice, which is known to increase with age [187,188]. This result is consistent with a previous observation that the increase of microgliosis that is observed in the cerebral cortex of 6-month-old WT mice can be attenuated by the neuronal expression of E2F4DN [12].

Making an effort to better understand the non-canonical functions and mechanisms of action of E2F4 will greatly benefit many fields, including the study of neuronal function and malfunction associated with neurodegenerative diseases and brain aging. Although the role of E2F4 as a repressor of the cell cycle has been extensively studied, and its mechanism is fairly known, being a critical molecule in the RB/E2F pathway, little is known about E2F4 implications in other cell processes. The latest studies challenge this paradigm, indicating that E2F4 has several roles in cells in addition to this regulatory function in the cell cycle. In this review, we have discussed a new perspective focusing on the regulation by E2F4 of various biological programs in the cell, regardless of its classical function. We have discussed the possible mechanisms that support these new roles, as well as the implications of these functions for disease research, including neurodegenerative diseases, and brain aging. The potential versatility of E2F4 is intriguing, but given that E2F4 is broadly expressed in the cell, can modulate the expression of a wide variety of genes, and can bind to various targets, many of which are involved in fundamental neuronal processes, it makes sense to investigate non-canonical functions and to include E2F4 as a key protein in different cellular and, particularly, neuronal functions. Understanding these non-canonical functions will likely reveal new insights into its role in controlling neuronal activity and associated diseases, which in turn could guide the development of new strategies to treat neurodegenerative diseases and brain aging. Therefore, E2F4 is a potential therapeutic target for diseases with cognitive impairment, such as AD. As an example of the potentiality of E2F4 as an intervention target, we have discussed a novel mouse model expressing a mutant form of E2F4 that has proven to be a multifactorial therapeutic molecule for AD and likely for other neurodegenerative conditions and brain aging.

## 8. Materials and Methods

### 8.1. Mice

C57BL6/J mice were purchased from ENVIGO (Indianapolis, Indiana, USA). Double transgenic mice in C57BL/6J genetic background expressing mutant human APP695 containing the Swedish (K670N, M671L), Florida (I716V), and London (V717I) familial AD (FAD) mutations, and human presenilin 1 harboring the M146L and L286V FAD mutations, under the control of the Thy1 promoter (Tg6799 or 5xFAD mice) were purchased from the Jackson Laboratory (Bar Harbor, Maine, USA) (strain #008730). The 5xFAD mice were genotyped as indicated by the Jackson Laboratory. Homozygous *Mapt*^tm1(EGFP)Klt^ KI mice expressing enhanced green fluorescent protein (EGFP) in neurons (EGFP mice) [185] were purchased from The Jackson Laboratory (strain #004779). This mouse strain was maintained on a mixed background of C57BL/6 and 129Sv or backcrossed to the C57BL/6 background. EGFP mice have a target mutation in the *Mapt* gene, characterized by the insertion of the coding sequence of EGFP into the first exon, thus disrupting the expression of the tau protein. This results in the neuron-specific expression of cytoplasmic EGFP. Tau is expressed at high levels in neurons [192], and homozygous mice mutants for tau are viable, fertile, and display no gross morphological abnormalities in the central or peripheral nervous systems [185]. Homozygous EGFP mice are viable, fertile, normal in size, and do not display any gross physical or behavioral abnormalities. The EGFP mice were genotyped as indicated by the Jackson Laboratory. These mice were used in this study as a control for E2F4DN mice. Homozygous EGFP mice were bred with hemizygous 5xFAD mice to generate littermates consisting of hemizygous EGFP mice with or without the 5xFAD transgene. *Mapt*^tm(mE2F4DN-myc)^ KI mice (E2F4DN mice) were generated following the procedure described by [12]. These mice express a dominant negative form of E2F4 equivalent to the mutant E2F4 used to prevent NT in chick neurons [13]. The KI strain was maintained on a mixed background of C57BL/6 and 129Sv or backcrossed to the C57BL/6 background. Homozygous E2F4DN mice were created by inbreeding mice containing one copy of the E2F4DN transgene. Homozygous E2F4DN mice are viable, fertile, normal in size, and do not display any gross physical or behavioral abnormalities, even though the tau protein has been deleted [12]. Homozygous E2F4DN mice were bred with hemizygous 5xFAD mice to generate littermates consisting of hemizygous E2F4DN mice with or without the 5xFAD transgene. Analyses were performed on hemizygous mice for both *Egfp* and *E2f4dn* transgenes to avoid the observed effects of a full *Mapt* null mutation in the phenotype of APP and APP/PS1 transgenic mice [193,194,195]. E2F4DN mice are available upon request for research purposes other than neurological, neurodegenerative, and aging diseases.

### 8.2. Antibodies

The mouse anti-NeuN mAb clone A60 (MAB377; Merck Millipore, Burlington, Massachusetts, USA) was used at a 1:1000 dilution for immunohistochemistry. The rabbit anti-GFAP pAb (ab7260, Abcam, Cambridge, UK) was diluted to 1:1000 for immunohistochemistry. Rabbit anti-Iba1 pAb (019-19741, Wako) was used at a 1:800 dilution for immunohistochemistry. The anti-E2F4 (Acetyl-Lys96) rabbit pAb (D12062, Assaybiotech, Fremont, CA, USA) was diluted to 1:1,100 for immunohistochemistry.

The donkey anti-rabbit IgG (H + L) highly cross-adsorbed secondary antibody Alexa Fluor 488 (Invitrogen, Waltham, MA, USA) was used at 1:1,000 dilution for immunohistochemistry. The goat anti-mouse IgG (H + L) cross-adsorbed secondary antibody Alexa Fluor 568 (Invitrogen, Waltham, MA, USA) was diluted 1:1000 for immunohistochemistry.

### 8.3. Tissue Processing

After anesthetizing the mice with intraperitoneal sodium pentobarbital (Dolethal; Vetoquinol, Alcobendas, Spain), administered at 50 mg/kg (body weight), they were transcardially perfused with phosphate buffered saline (PBS), and then with 4% paraformaldehyde (PFA). Brains were finally postfixed overnight at 4 °C with 4% PFA and cryoprotected by sinking in 32% sucrose in PBS at 4 °C. The brains were then embedded in 3% agarose gels prepared in 0.1 phosphate buffer, pH 7.37, before cutting them with a vibratome (50 μm). Vibratome sections were then stored at −20 °C in a solution of 3% glycerol (Panreac, San Fernando de Henares, Spain)/3% ethylene glycol (Panreac) prepared in 100 mM phosphate buffer, pH 7.37.

### 8.4. Immunohistochemistry

The vibratome sections were permeabilized and blocked in 0.4% Triton X-100 in PBS (PBTx) containing 10% fetal calf serum (FCS) for 3 h. They were then incubated overnight at 4 °C with the primary antibodies in 0.1% PBTx containing 1% FCS. After five washes of 20 min with 0.1% PBTx, the sections were incubated with the secondary antibodies plus 100 ng/mL DAPI in 0.1% PBTx for 3 h at room temperature. The sections were then washed five times with 0.1% PBTx, and mounted with ImmunoSelect antifading mounting medium DAPI (CliniSciences, Nanterre, France).

### 8.5. Quenching of Lipofuscin Autofluorescence Signal

Lipofuscin was quenched with TrueBlackTM Lipofuscin Autofluorescence Quencher (Biotium, Fremont, CA, USA). Briefly, the vibratome sections were washed once with PBS and treated for 30 s with TrueBlack 1× prepared in 70% ethanol. Finally, the sections were washed three times with PBS, and then immunostained as described above.

### 8.6. Confocal Microscopy and Image Analysis

Confocal images were acquired at 20× magnification with a Leica SP5 confocal microscope. Image analysis was performed using ImageJ (Fiji). The images used for the analysis (at least two mosaic images per tissue and animal) were maximum intensity projections created as output images whose pixels corresponded to the maximum value of each pixel position (in xy) across all stack images (z). DAPI staining was used to define the cortical layers and hippocampal structures. In order to analyze the area occupied by GFAP and Iba1, a threshold was set to highlight the area to be quantified. Quantification of the area occupied by Iba1-labeled microglia was achieved using a multi-step algorithm. First, Iba1-labeled microglia were segmented by applying a gray-scale attribute opening filter (area minimum: 25 pixels; connectivity: 8) to an 8-bit maximum projection. An opening morphological filter (1-pixel radius octagon) was then used effectively to separate microglia soma from processes before a maximum entropy threshold was used to discriminate microglial cells or astrocytes from the image background.

### 8.7. Statistical Analysis

The quantitative data are represented as the mean ± s.e.m. Two-way ANOVA analysis was performed for the quantitative analysis of immune cells, followed by a post hoc Newman–Keuls test.

### 8.8. Gene Ontology Analysis

Gene ontology analyses (both CC and BP ontology) were performed using the database for annotation, visualization, and integrated discovery (DAVID) software [196,197] “https://david.ncifcrf.gov/ (accessed on 9 May 2022)”.

## Figures and Tables

**Figure 1 ijms-23-12093-f001:**
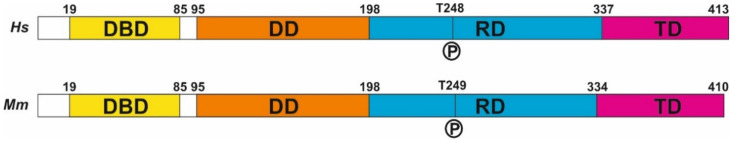
E2F4 structure. The structure of human (*Hs*) and mouse (*Mm*) E2F4, derived from NCBI Reference Sequences NP_001941.2 (*H. sapiens*) and NP_683754.1 (*M. musculus*). DBD: DNA binding domain, DD: dimerization domain, RD: regulatory domain, TD: transactivation domain.

**Figure 2 ijms-23-12093-f002:**
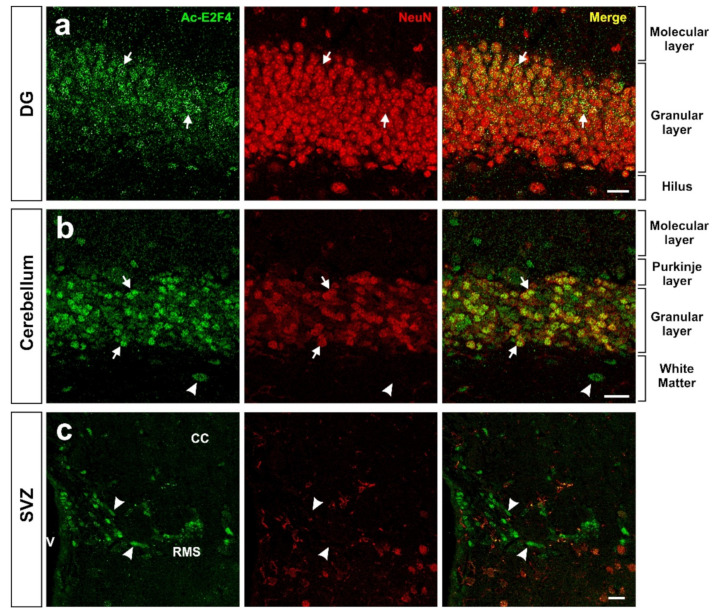
Expression pattern of acetylated E2F4 in the dentate gyrus (DG) (**a**), cerebellum (**b**), and subventricular zone (SVZ) (**c**) of 2.5-month-old WT mice. One single confocal plane showing co-immunostaining with anti-acetylated E2F4 (Ac-E2F4) and anti-NeuN (NeuN) antibodies in sections from the indicated brain areas. NeuN specifically labels neurons. Ac-E2F4 immunostaining in NeuN-positive (arrows) and NeuN-negative (arrowheads) cells is shown. V: ventricle; CC: corpus callosum; RMS: rostral migratory stream. Scale bar: 20 μm.

**Figure 3 ijms-23-12093-f003:**
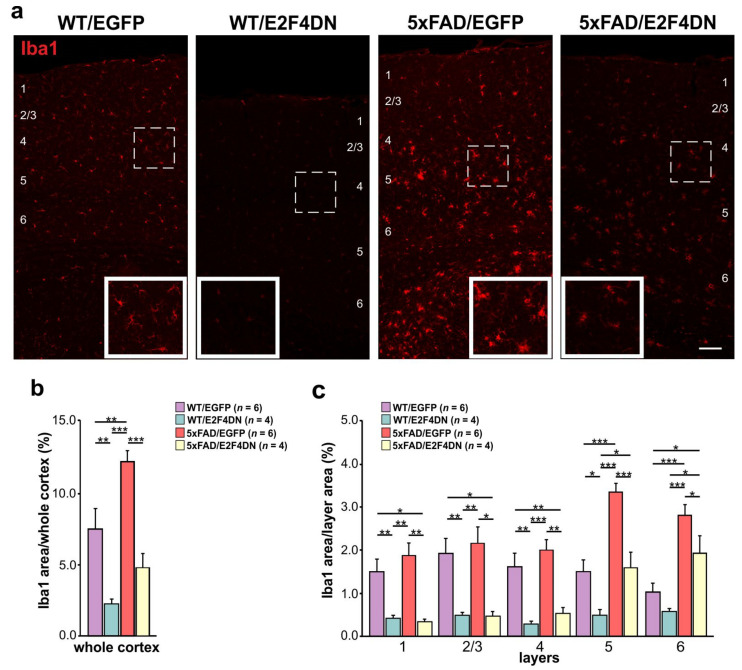
Modulation of microgliosis by E2F4DN in the cerebral cortex of 1-year-old 5xFAD mice. (**a**) Iba1 immunostaining in the cerebral cortex of mice of the indicated genotypes at 1 year. Numbers refer to the different cortical layers (identified by DAPI staining; not shown). Inserts show the high magnifications of the indicated dashed boxes. (**b**) Percentage of the area occupied by Iba1 immunostaining in the cerebral cortex of mice of the indicated genotypes at 1 year. (**c**) Percentages of the area occupied by Iba1 immunostaining in the indicated cortical layers at 1 year. * *p* < 0.05; ** *p* < 0.01; *** *p* < 0.001 (one-way ANOVA followed by post hoc Newman-Keuls test). Scale bar: 100 μm.

**Figure 4 ijms-23-12093-f004:**
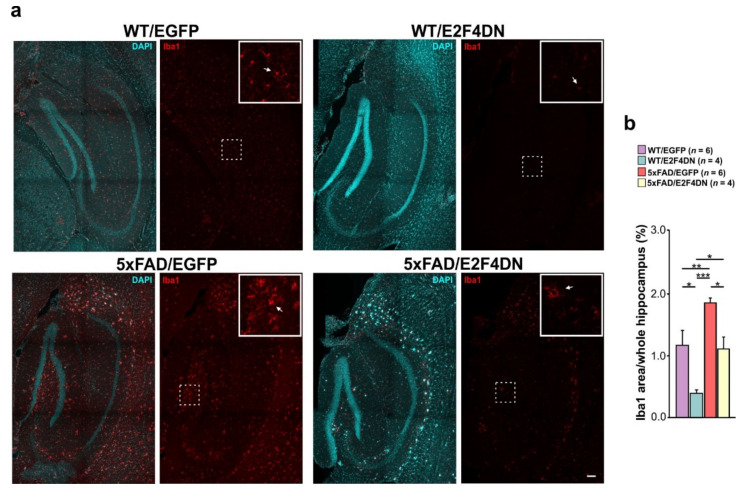
Attenuation of microgliosis by E2F4DN in the hippocampus of 1-year-old 5xFAD mice. (**a**) Iba1 immunostaining (arrow) in the hippocampus of mice with the indicated genotypes. Inserts show the high magnifications of the indicated dashed boxes. DAPI counterstaining was included to identify the hippocampus structure. (**b**) Percentage of the area occupied by Iba1 immunostaining in the hippocampus. * *p* < 0.05; ** *p* < 0.01; *** *p* < 0.001 (one-way ANOVA, followed by post hoc Newman-Keuls). Scale bar: 100 μm.

**Figure 5 ijms-23-12093-f005:**
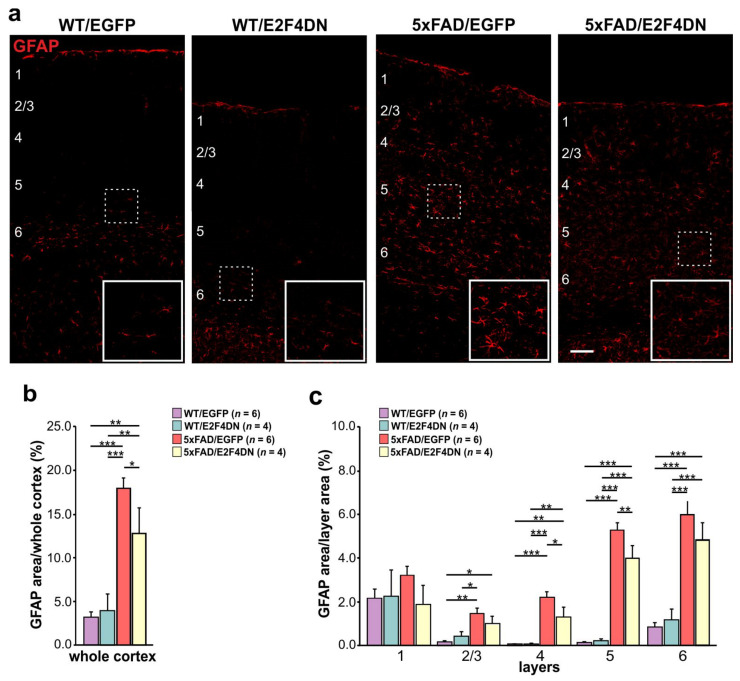
Modulation of astrogliosis by E2F4DN in the cerebral cortex of 1-year-old 5xFAD mice. (**a**) GFAP immunostaining in the cerebral cortex of mice with the indicated genotypes. Numbers refer to the different cortical layers (identified by DAPI staining; not shown). Inserts show the high magnifications of the indicated dashed boxes. (**b**) Percentage of the area occupied by GFAP immunostaining in the cerebral cortex. (**c**) Percentages of the area occupied by GFAP immunostaining in the cortical layers. * *p* < 0.05; ** *p* < 0.01; *** *p* < 0.001 (one-way ANOVA followed by post hoc Newman-Keuls test). Scale bar: 100 μm.

**Figure 6 ijms-23-12093-f006:**
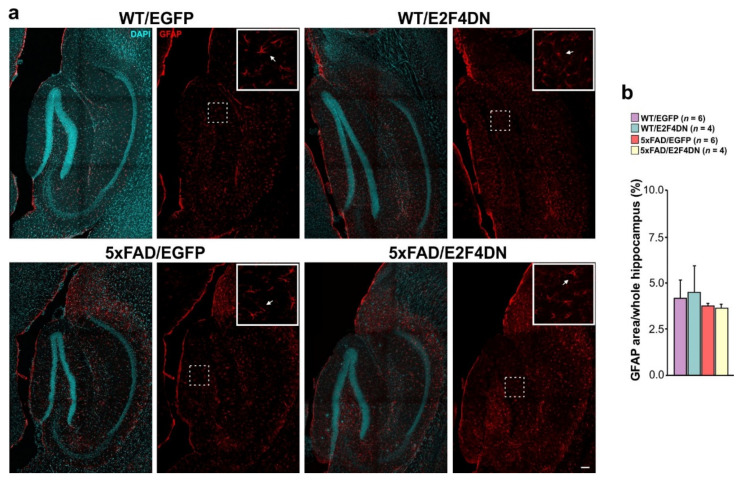
Modulation of astrogliosis by E2F4DN in the hippocampus of 1-year-old 5xFAD mice. (**a**) GFAP immunostaining (arrow) in the hippocampus of mice with the indicated genotypes. Inserts show the high magnifications of the indicated dashed boxes. DAPI counterstaining was included to identify the hippocampus. (**b**) Percentage of the area occupied by GFAP immunostaining in the hippocampus. Scale bar: 100 μm.

## Data Availability

Data generated or analyzed during this study are included in this published article and its Appendix A.

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
