# Peer review of "E2F4DN Transgenic Mice: A Tool for the Evaluation of E2F4 as a Therapeutic Target in Neuropathology and Brain Aging"

_ijms, 2022, doi:10.3390/ijms232012093_

Round 1
Reviewer 1 Report
This review article compiles a lot of literature on E2F4. However, the subject is E2F4DN transgenic mice, so it would be better to have more literature describing such transgenic mice. Perhaps it can be presented in tabular form or otherwise.
Other minor issues such as:
1. The quality of Figure 2 can be made clearer.
2. What is the reason for looking at these brain regions? Are they different? It seems different from NeuN merge results
3. Why is there no difference in astrogliosis in 1-year-old 5XFAD mice? This needs to be discussed.
4. The brain aging issue does not seem to be described.
Author Response
This review article compiles a lot of literature on E2F4. However, the subject is E2F4DN transgenic mice, so it would be better to have more literature describing such transgenic mice. Perhaps it can be presented in tabular form or otherwise.
This Ms. is mainly focused on the role of E2F4 as a therapeutic target in neuropathology and brain aging. For the evaluation of this role, we introduce E2F4DN transgenic mice in the context of a special issue on “Transgenic Mice in Human Diseases: Insights from Molecular Research 3.0”. Since the E2F4DN transgenic mice constitute a novel strain generated by our group, there are only a few papers in the whole literature describing these mice. All these papers were included in the former version of the Ms.:
“Mapt tm(mE2F4DN-myc) KI mice (E2F4DN mice) were generated following the procedure described by [12]. These mice express a dominant negative form of E2F4 equivalent to the mutant E2F4 used to prevent NT in chick neurons [13].” (lines 720-723 from the previous version of the Ms.).
There was also information about these mice in reference 117. We have clarified this issue by inserting the following text in the revised version of the Ms.:
“[...] while the expression of E2F4DN prevents these latter effects [11,12,117].” (lines 484-485).
Finally, E2F4DN transgenic mice were generated using the same strategy as that used by Tucker et al (2001) to generate the EGFP transgenic mice. This was indicated in the main text:
“As a control we used KI mice expressing EGFP under the Mapt promoter (EGFP mice) [185].” (lines 542-543 from the previous version of the Ms.).
Other minor issues such as:
- The quality of Figure 2 can be made clearer.
The previous version of Figure 2 was based on the projection of several confocal sections. Figure 2 now contains only one confocal section per panel with higher magnification than in the former figure. Figure 2 legend has also been modified:
“Expression pattern of acetylated E2F4 in the indicated brain regions of 2.5 month-old WT mice. One single confocal plane showing co-immunostaining with anti-acetylated E2F4 (Ac-E2F4) and anti-NeuN (NeuN) antibodies in sections from the indicated brain areas. NeuN specifically labels neurons. Ac-E2F4 immunostaining in NeuN-positive (arrows) and NeuN-negative (arrowheads) cells is shown. DG: dentate gyrus; SVZ: subventricular zone; V: ventricle; CC: corpus callosum; RMS: rostral migratory stream. Scale bar: 20 μm.” (lines 616-621).
- What is the reason for looking at these brain regions? Are they different?
These are the only structures where we could find acetylated-E2F4-specific immunolabeling. This is now indicated in the revised version of the Ms.:
“Using an acetylated K96-specific antibody, we have verified that K96 becomes acetylated in some structures of the adult mouse brain in vivo (Fig. 2).” (lines 163-164)
It seems different from NeuN merge results
Yes, this was evident in the rostral migratory stream, and was actually pointed out in the former version of the Ms.:
“[…] in NeuN-negative cells located in the rostral migratory stream (RMS) (Fig. 2c), likely being neural progenitors.” (lines 166-167 in the former version).
The revised version of Figure 2 now contains arrows to label acetylated E2F4 immunostaining in NeuN-positive cells. In addition, acetylated E2F4 immunostaining in NeuN-negative cells has been labeled with arrowheads.
In addition, we also identified in the cerebellum NeuN-negative cells acetylated E2F4-specific immunoreactivity. This information has been included in the revised version of the Ms.:
“Some NeuN-negative cells in the cerebellum also showed acetylated E2F4-specific immunoreactivity (Fig. 2b).” (lines 168-169).
- Why is there no difference in astrogliosis in 1-year-old 5XFAD mice? This needs to be discussed.
Our results demonstrate that the presence of neuronal E2F4DN expression results in less astrogliosis in the cerebral cortex of 1 year-old 5xFAD mice. This was already indicated in the former version of the Ms.:
“As observed at 3 months [12], the presence of E2F4DN significantly diminished the area occupied by GFAP immunoreactivity in 1 year-old 5xFAD mice (Fig. 5a,b), further supporting the hypothesis that neuronal E2F4DN expression attenuates the reactive astrogliosis observed in 5xFAD mice.” (lines 592-595 in the former version of the Ms.).
In contrast, we agree that no increase of GFAP was detected in the hippocampus, likely because astrocytes already express GFAP in this brain region. This was indicated in the previous version of the Ms.:
“In the hippocampus, where GFAP is expressed by astrocytes at high basal levels in both WT and 5xFAD mice (Fig. 6a), no difference was observed when E2F4DN was expressed in both WT/E2F4DN and 5xFAD/E2F4DN mice (Fig. 6a,b).” (see lines 601-603 in the former version of the Ms.).
Nevertheless, we now stress this point in the revised version of the Ms., by including the following text in the Discussion:
“Our results confirm the capacity of E2F4DN to attenuate microgliosis in the cerebral cortex and hippocampus of 5xFAD mice even after one year, thus indicating that it has long-lasting therapeutic effects. In addition, E2F4DN is able to decrease the area occupied by GFAP cells (i.e. reactive astrocytes) in the cerebral cortex of 5xFAD mice, while no changes in the area occupied by GFAP were observed in the hippocampus. This latter result contrasts with the observation that the area occupied by GFAP in the hippocampus of 3 month-old 5xFAD mice is decreased in the presence of neuronal expression of E2F4DN [12]. This discrepancy may be explained by the attenuation of astrocytosis in the hippocampus of 5xFAD mice at 1 year of age, a tissue where, in contrast to cerebral cortex, GFAP is already expressed by non-reactive astrocytes.” (lines 670-679).
- The brain aging issue does not seem to be described.
The former version of the Ms. included the following text:
“[…] this same effect was also observed in WT mice expressing E2F4DN (Fig. 4), further supporting the hypothesis that the neuronal expression of our molecule can reverse the increase of microgliosis associated with brain aging in the hippocampus [187-189].” (lines 575-578 from the previous version of the Ms.)
The revised version of the Ms. now contains a paragraph in the Discussion section addressing this issue:
“Our results are also in favor of the hypothesis that E2F4DN has a role in preventing brain aging. This is evidenced by the capacity of our therapeutic protein to reduce the levels of microgliosis in the cerebral cortex and hippocampus of 1 year-old WT mice, which is known to become increased with age [187,188]. This result is consistent with previous observation that the increase of microgliosis that is observed in the cerebral cortex of 6 month-old WT mice can be attenuated by the neuronal expression of E2F4DN [12].” (lines 685-690).
Reviewer 2 Report
This manuscript combines the experimental part with a significant review on the role of the transcription factor E2F4 in brain aging and the development of neurodegenerative processes. The issues discussed are indeed quite interesting, perhaps controversial in some aspects, but that is why they are important for publication. But it seems to me that it would be better to divide the article into an experimental, albeit not very large article, and then a full-fledged review article.
Author Response
We feel that the amount and novelty of experimental work included in our Ms. does not justify a full experimental paper. This is even worsened by the fact that the experimental part of the Ms. is heterogeneous, which impedes to delineate a consistent story in the absence of the argumentation and studies included in the review part. The Ms. includes a figure illustrating the presence of acetylated E2F4 in some brain regions, which fits with the part focused on covalent modifications of E2F4. In addition, it includes another set of figures illustrating how the expression of E2F4DN reduces the neuroinflammatory phenotype of 5xFAD mice at 1 year of age. These latter results complement a previous paper from our lab showing that E2F4DN attenuates the neuroinflammatory phenotype of 3 and 6 month-old 5xFAD mice. Therefore, the 1 year results cannot be published as the major point of a full paper due to their reduced novelty. In contrast, they deserve to be published in the review as a complement of the previous paper from our lab. We believe that an article combining “the experimental part with a significant review on the role of the transcription factor E2F4 in brain aging and the development of neurodegenerative processes” fits quite well with a special issue on “Transgenic Mice in Human Diseases: Insights from Molecular Research 3.0”.